# Virus diversity, wildlife-domestic animal circulation and potential zoonotic viruses of small mammals, pangolins and zoo animals

Xinyuan Cui[1,20], Kewei Fan[2,20], Xianghui Liang[1,20], Wenjie Gong[3,4,20], Wu Chen[5,20], Biao He[4,20], Xiaoyuan Chen[1], Hai Wang[1], Xiao Wang[1], Ping Zhang[1], Xingbang Lu[1], Rujian Chen[1], Kaixiong Lin[6], Jiameng Liu[1], Junqiong Zhai[5], Ding Xiang Liu[7,8], Fen Shan[5], Yuqi Li[2], Rui Ai Chen[8], Huifang Meng[1], Xiaobing Li[1,2], Shijiang Mi[3,4], Jianfeng Jiang[3,4], Niu Zhou[5], Zujin Chen[5], Jie-Jian Zou[9], Deyan Ge[10], Qisen Yang[10], Kai He[11], Tengteng Chen[6], Ya-Jiang Wu[5], Haoran Lu[12], David M. Irwin[13,14], Xuejuan Shen[1], Yuanjia Hu[1], Xiaoman Lu[1], Chan Ding[15,16] ✉, Yi Guan[17,18] ✉, Changchun Tu[4,16] ✉ & Yongyi Shen[1,19] ✉

Wildlife is reservoir of emerging viruses. Here we identified 27 families of mammalian viruses from 1981 wild animals and 194 zoo animals collected from south China between 2015 and 2022, isolated and characterized the pathogenicity of eight viruses. Bats harbor high diversity of coronaviruses, picornaviruses and astroviruses, and a potentially novel genus of *Bornaviridae*. In addition to the reported SARSr-CoV-2 and HKU4-CoV-like viruses, picornavirus and respiroviruses also likely circulate between bats and pangolins. Pikas harbor a new clade of Embecovirus and a new genus of arenaviruses. Further, the potential cross-species transmission of RNA viruses (paramyxovirus and astrovirus) and DNA viruses (pseudorabies virus, porcine circovirus 2, porcine circovirus 3 and parvovirus) between wildlife and domestic animals was identified, complicating wildlife protection and the prevention and control of these diseases in domestic animals. This study provides a nuanced view of the frequency of host-jumping events, as well as assessments of zoonotic risk.

The emergence of COVID-19 and the recent multi-country mpox outbreak in non-endemic countries highlights the ongoing threat of the emerging zoonotic infectious diseases. Wildlife are important microbial reservoirs and responsible for ~70% of emerging infectious diseases[1]. Accordingly, the spillover of wildlife viruses poses an ongoing challenge to human and animal health[2].

Small wild mammals, such as bats and rodents, are the most important known natural hosts of zoonotic viruses[3]. A wide variety of viruses, such as severe acute respiratory syndrome coronavirus 2 (SARS-CoV-2), SARS-CoV, Middle East respiratory syndrome coronavirus (MERS-CoV), henipaviruses, and filoviruses originated from bat viruses and cause severe epidemic or endemic diseases in humans[4–6]. Indeed, it is estimated that perhaps 66,000 people are silently infected with previously unknown bat coronaviruses annually[7]. Rodents are also well-known hosts of many human pathogens, including hantaviruses and mammarenaviruses, and are the likely hosts of human coronaviruses OC43 and HKU1, which have had a significant impact on public health[8,9].

Until the emergence of COVID-19, pangolins were largely neglected as potential hosts for viruses, although they have been identified as carrying a number of SARS-CoV-2-related and MERS-CoV-related

viruses[10–13]. Since this time, many additional pathogens, such as human parainfluenza 3, human respiratory syncytial virus, and murine respirovirus have been identified in these endangered animals[14–18]. Similarly, zoo animals are kept at very high densities and in close contact with humans and domestic animals, providing opportunities for both zoonosis and reverse-zoonosis[19]. For example, during the COVID-19 pandemic, there have been multiple transmissions of SARS-CoV-2 from human to zoo animals, such as pumas, tigers, lions, and gorillas[20–22].

There has been a growing effort to employ metagenomic approaches to document the broad spectrum of pathogen diversity of wildlife species[23, 24]. Given the huge diversity of viruses, and the increasing contact frequency (both direct and indirect) at the human-domestic animal-wildlife interface, it is reasonable to speculate that there are a multitude of unrecorded viruses and undocumented cross-species transmission events. In addition, most current surveillance exercises rely on metagenomic sequencing alone, with limited attempts at virus isolation and animal infection experiments. Clearly, a better understanding of viruses involved in spillover and successful host-jumping events will greatly enhance our understanding of the drivers of disease emergence.

Herein, we used meta-transcriptomic sequencing to determine the viromes in small mammals (bats, rodents, insectivores, and pikas), pangolins and zoo animals collected in South and Southwest China (Supplementary Data 1), and from this determine the extent and pattern of cross-species virus transmission. We also isolated some of the viruses identified and performed experimental infection studies. This work documents virus diversity and identifies viruses in wildlife with zoonotic potential.

## Results

### Overview of animal viromes
A total of 503 libraries representing 2175 individual animals that were collected between 2015 and 2022 were sequenced (Supplementary Data 1). In brief, there were 214, 123, 18, 21, 56, and 71 libraries from bats, rodents, pikas, insectivores, pangolins and zoo animals, respectively. An average of 12 Gb of sequence data was obtained for each library. We focus only on the viruses that are able to infect vertebrates, while those infecting archaea, bacteria, fungi, invertebrates, and plants were excluded. An overview of the reads for the mammalian viruses is presented in Fig. 1, Supplementary Fig. 1, and Supplementary Data 2. A total of 328 viruses were identified through phylogenetic analyses, with 171 of them having near-complete genomes, and 167 of them were unreported (Supplementary Data 3 and Supplementary Table 1).

Rodents had 20 virus families, followed by bats, insectivores and zoo animals (with 19, 15, and 14 virus families, respectively), whereas pikas and pangolins showed the presence of the fewest number of viral families (nine each). Viral reads from the families *Arenaviridae*, *Arteriviridae*, *Astroviridae*, *Caliciviridae*, *Circoviridae*, *Coronaviridae*, *Flaviviridae*, *Hantaviridae*, *Hepeviridae*, *Herpesviridae*, *Paramyxoviridae*, *Parvoviridae*, *Picornaviridae*, and *Reoviridae* were widely distributed in these animals. These viruses were further confirmed by RT-PCR (RNA viruses) and PCR (DNA viruses), with the details shown in Supplementary Data 4. The reads of *Retroviridae* and *Herpesviridae* derived from the host genomes could not be excluded since the genomes of most of the tested animals are not available. Therefore, the abundance of *Retroviridae* and *Herpesviridae* in the heatmap might be overestimated.

### Diversity and genomic organization of coronaviruses
The reads of coronaviruses (CoVs) were detected in 49, 11, 7, and 9 libraries from bats, rodents, pangolins, and pikas, respectively. Notably, seven bat-derived betacoronaviruses (β-CoVs) identified in genus *Rhinolophus* had a close relationship with SARS-CoV. A β-CoV (bat GZ/L165.18/2022) was detected in a great evening bat (*Ia io*) in the family

Vespertilionidae. It had 98.1% and 74.5% genomic sequence identity and 99.8% and 94.8% RdRp amino acid identity with bat MERS-CoV-related (accession number MG021452) and human MERS-CoV (accession number NC_019843) respectively. In total, 19 bat α-CoVs were detected in the bats *Hipposideros pomona*, *Hipposideros cf. larvatus*, *Rhinolophus pusillus*, *Rhinolophus affinis*, *Miniopterus pusillus*, *Miniopterus schreibersii*, *Myotis davidii*, *Myotis chinensis*, and *Myotis ricketti*. Recombination events were detected in the genomes of these CoVs (Supplementary Data 5, and Supplementary Fig. 2). Of note was the swine acute diarrhea syndrome coronavirus-related (SADSr-CoV) detected in *Rhinolophus pusillus* (bat CoV GD/L31.18/2021, collected in Guangdong province in 2021). Its genome exhibited 82.8% nucleotide identity with the SADS-CoV genome sequence.

For rodents, two bamboo rat CoVs clustered with the murine hepatitis virus, a rodent CoV identified in *Niviventer confucianus* grouped with *Apodemus peninsulae* CoV and a bat CoV (green box in Fig. 2a). All of these are Embecovirus. In contrast, *Eothenomys eva* CoV clusters with a rodent and a pika CoVs and belong to the α-CoVs (yellow box in Fig. 2a).

Notably, three β-CoVs were detected in samples from Malayan pangolins that were from the same batch of custom confiscated animals used in our previous study[10]. These viruses had 99.6–99.9% nucleotide identity with the pangolin SARSr-CoV-2 (GD1)[10], and also have a close relationship with SARS-CoV-2. Other pangolins collected in Guangdong and Guangxi provinces did not carry this virus. A bat HKU4-CoV-like virus (β-CoV lineage C) was detected in a Malayan pangolin collected in Guangxi Province in 2020 with 98.7% and 86.9% genome sequence, and 99.7% and 96.6% RdRp amino acid identities to pangolin CoV P252T and bat HKU4-CoV (red box in Fig. 2a).

RT-PCR further revealed that rectal samples from 15 of 53 pikas collected in three different regions (Mianyang, Aba, and Ganzi) of the Tibetan plateau were positive for CoV viruses. Six near-complete CoV genomes were assembled that formed a sister clade of Embecovirus (Fig. 2a). These six pika-derived CoV genomes exhibited 95.9–99.9% nucleotide identity among themselves and had 47.3–64.2% nucleotide identity with other viruses of the subgenus Embecovirus. Interestingly, their genome organization differed from those of other CoVs, with the hemagglutinin-esterase (*HE*) gene being 3′ of the S gene, whereas in the mouse hepatitis virus (MHV), rabbit HKU14-CoV, and human HKU1-CoV, this gene is 5′ of the *S* ORF (Fig. 2b). We failed to detect any recombination events in pika CoVs, and simplot analysis showed that they displayed high distance from other viruses of the Embecovirus across the whole-genome sequence (Fig. 2b).

### Phylogeny and pathogenicity of the *Flaviviridae*
Sequence reads for flaviviruses were detected in 15 of the 214 bat libraries, 23 of the 123 rodent libraries, 34 of the 56 pangolin libraries and four of the 71 the zoo animal libraries (Supplementary Data 2). Assembles that had near-complete *NS5* gene sequences were used to estimate a phylogenetic tree (Fig. 3a). Two bat and seven rodent flaviviruses belonged to the genus *Pegivirus*, whereas two other bat and three additional rodent viruses fell in the genus *Hepacivirus*. All of the pangolin and one bamboo rat flaviviruses were pestiviruses (PeVs). Interestingly, the pangolin pestivirus was detected in the lung (10 of 14), intestine (2 of 3), liver (4 of 8), kidney (3 of 6), spleen (6 of 10), placenta (1 of 1), and muscle (7 of 11). The identities of the nucleotide and amino acid sequences of the *NS5* gene of the pangolin PeVs vary greatly between individuals (76.7–83.5% and 84.8–89.0%), with the exception of 100% sequence identity between the PeVs in individuals P21 and P22. Sequence identity analysis of seven complete genomes from different tissues of a pregnant pangolin (M5) and its fetus (M5 fetus) showed that their genomes were 99.8% identical, suggesting that they were infected with the same PeV strain.

Phylogenetic analysis revealed that all of the pangolin PeVs (including the DYPV strain identified in pangolins collected in

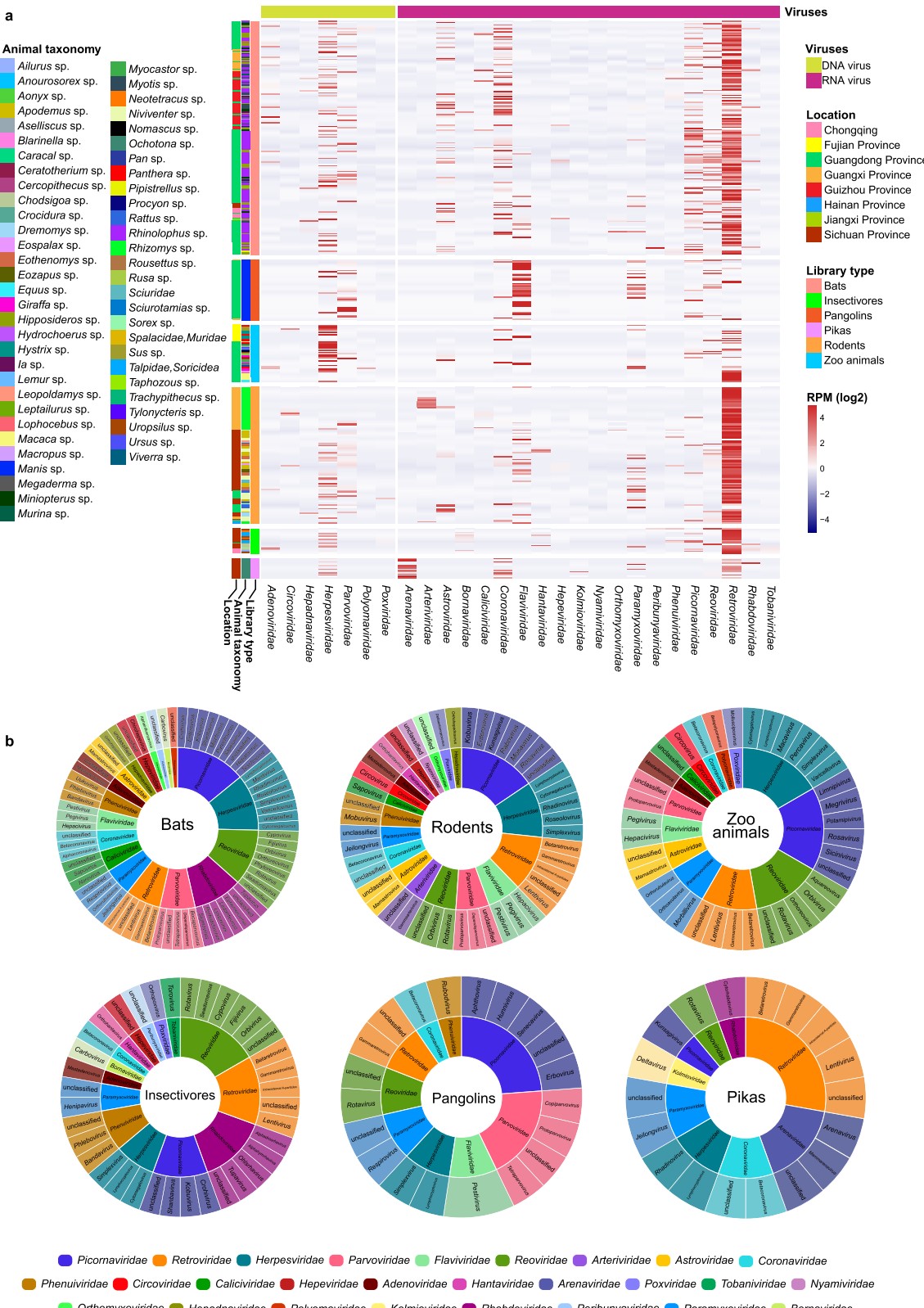

**Fig. 1 | Overall view of the viral reads in the meta-transcriptomic data.**
**a** Heatmap based on normalized sequence reads of 27 families of mammalian viruses in each pooled sample and their Chinese province of sampling. Names of mammalian viral families are presented in the text row at the bottom. **b** Overview of virus classifications, from family to genus, of the viruses identified in bats, rodents, pikas, insectivores, pangolins, and zoo animals in this study. Different families are labeled in different colors.

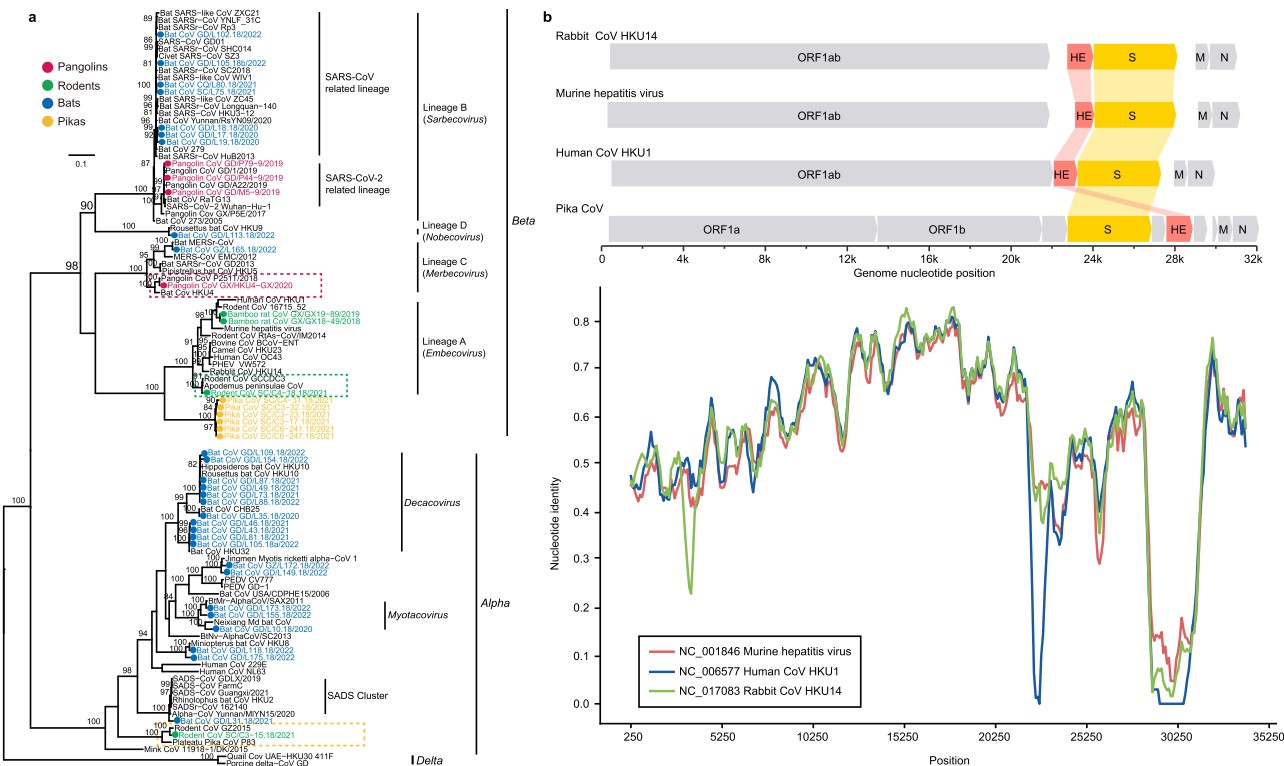

**Fig. 2 | Phylogenetic diversity of the coronaviruses identified here. a** Phylogeny of the *Coronaviridae* based on amino acid sequences of the *RdRp* gene. Viruses identified in this study are color-coded according to the animal from which they were sampled. The scale bar depicts the number of amino acid substitutions per site. The tree was rooted with δ-CoVs. **b** Comparison of the genomic organization and similarity plot of Pika-CoV and related viruses. Predicted ORFs are indicated. Parameters for the similarity plots are: window, 500 bp; step, 100 bp.

Zhejiang province, China, 81.0–91.6% amino acid identity) formed an independent clade, which is sister to the artiodactyl lineage of pestiviruses. The amino acid sequences encoded by these viruses had ~50% amino acid identity with the classical swine fever virus (Gen-Bank: AF326963) (Fig. 3b).

One variant of PeVs from pangolin individual P42 was isolated and named pangolin pestivirus/GD/P42 (Fig. 3c). To further characterize its potential host range and threat to domestic animal industry, we conducted experimental infections in mice, rats, guinea pigs, rabbits and pigs. At 7 days-post-infection (DPI), all inoculated animals were euthanized, and their lung, kidney, spleen, and lymph nodes (pigs and rabbits only) were collected. According to RT-PCR assay, all tissues from these animals were negative for this virus. To further examine its potential ability to infect cow and human, MDBK (cow) and A549 (human) cell lines were chosen. Again, we failed to detect replication of this virus in either cell line.

### Diversity of other RNA viruses

Sequence reads for picornaviruses (PicoVs) were identified in 65, 5, 10, and 5 libraries from bats, rodents, pangolins, and insectivores, respectively (Supplementary Data 2). Phylogenetic analysis, based on RdRp showed that all of the insectivore and some of the bat PicoVs form an independent lineage (Fig. 4a and Supplementary Fig. 3a). Bats harbor the most diverse PicoVs, with 44 viral sequences, 10 of which were assigned to a previously reported Bat PicoV 3 (NC_015934), and all with very high amino acid identity (94.5–95.4%). Interestingly, eight of the bat PicoVs cluster with a pangolin PicoV, with 84.8–95.8% amino acid identity (blue box, Fig. 4a and Supplementary Fig. 3a). Two of the pangolin PicoVs (GD/P47-7/2019 and GX/HKU4−GX/2020) cluster with a known Pangolin hunnivirus, and two of the other pangolin PicoVs (GD/P70-2/2019 and GD/M5-2/2019) form an independent lineage that is a sister clade of Senecavirus A. Two bat PicoVs

(GD/L24.18b/2020 and GD/L25.18b/2020) have 89.8% amino acid identity with a bamboo rat PicoV (GX/B45M0208/2019). They are the sister clade of human Aichivirus A, with 89.6% and 86.6% amino acid identity.

Sequence reads for pangolin respiroviruses were identified in 15 of the 56 pangolin libraries (Supplementary Data 2). The *L* genes of these pangolin viruses and two bat-derived viruses (L86 and L84) were identical to our previously reported pangolin respirovirus M5 (red box, Fig. 4b and Supplementary Fig. 3b). The amino acid sequence of the *L* gene of the pika virus is identical with that of a Chinese jumping mouse (*Eozapus setchuanus*) which was collected in the same region (yellow box, Fig. 4b and Supplementary Fig. 3b). Three respiroviruses were identified in South China tigers, two of which exhibit 99.5% and 99.3% amino acid identity with the parainfluenza virus 5 isolated from the lesser panda (KX100034) and canine (KP893891), as well as 99.3% amino acid identity with the pig orthorubulavirus 5. The third respirovirus clustered with the cat morbillivirus at 87.9% amino acid identity (purple box, Fig. 4b and Supplementary Fig. 3b).

Astroviruses were identified in 42 of the 214 bat libraries, 9 of the 123 rodent libraries, and 2 of the 71 zoo libraries. A total of 27 representative contigs encoding the capsid protein precursor were used in phylogenetic analysis (Fig. 4c and Supplementary Fig. 3c). Bats had the most diverse astroviruses, indicating that they are an important natural reservoir, while the astroviruses in caracals have 98.0% amino acid identity with cat mamastrovirus 2 (MH253864) (marked in purple box, Fig. 4c and Supplementary Fig. 3c), suggesting potential circulation of this virus between domestic and wild felids.

A new clade of arenaviruses (AreV) was detected from 11 pika libraries representing lung, kidney, and rectum tissues. These sequences form a sister clade to mammarenaviruses (Fig. 4d and Supplementary Fig. 3d), yet with only 21.3−26.9% amino acid identity in

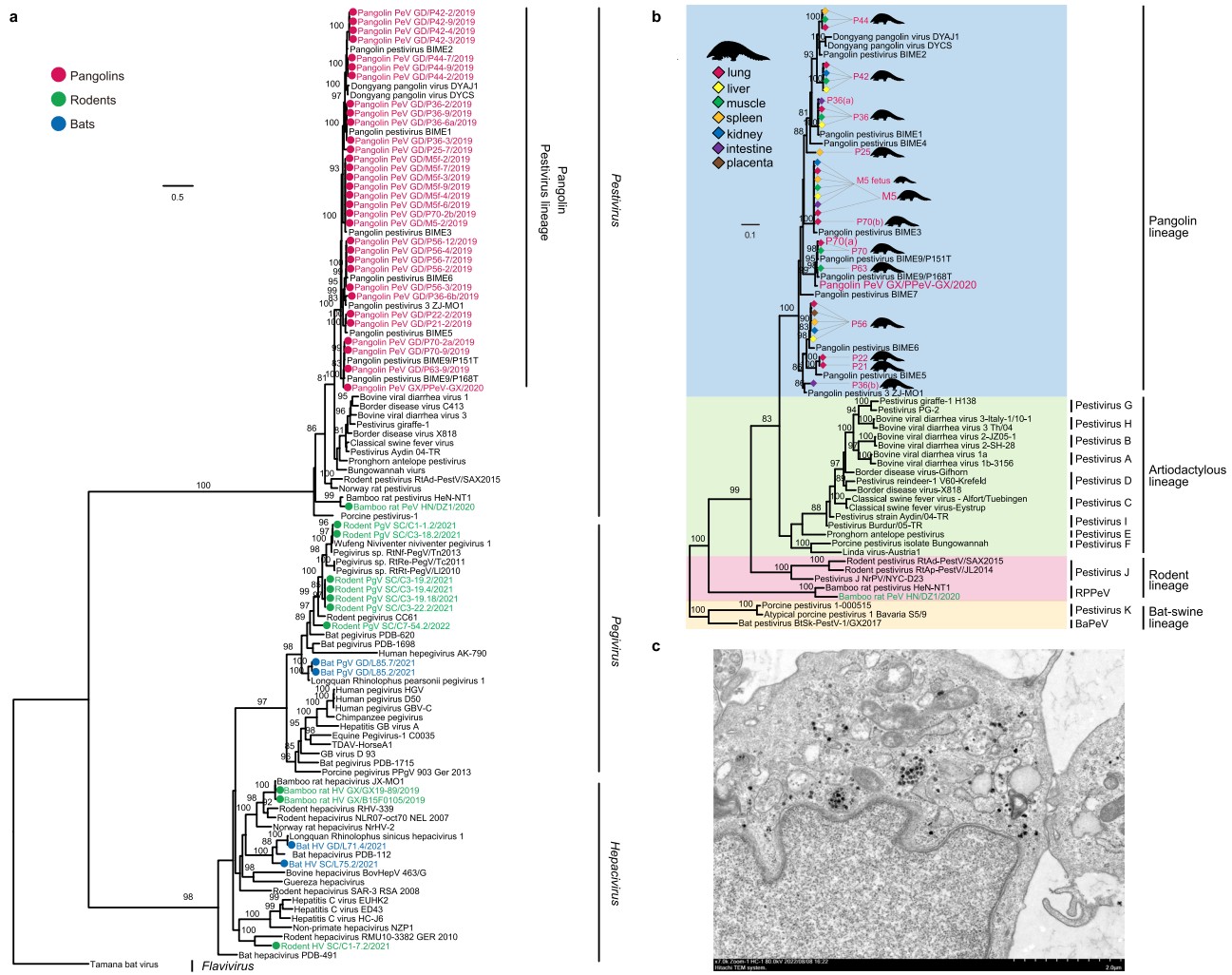

**Fig. 3 | Phylogenetic diversity of flaviviruses identified here. a** Phylogeny of the *Flaviviridae* based on amino acid sequences of the *NS5* gene. Viruses identified in this study are color-coded according to the animal from which they were sampled. The tree was rooted with Tamana bat virus. Scale bar depicts the number of amino acid substitutions per site. **b** Phylogeny of pestiviruses based on the amino acid sequences of the *NS5* gene. Representatives of Pestivirus A-K are incorporated. The tree was rooted with Pestivirus K and BaPeV. Viruses detected in the different tissues of the pangolins are labeled in different colors. **c** Viral particles of pangolin pestivirus, grown in Vero E6 cell culture, seen by transmission electron microscopy image. Scale bar, 2 μm. The experiment was performed independently in triplicate with similar results.

the *L* gene. Because they are so divergent, we propose that the pika clade be assigned as a new genus within the *Arenaviridae*. These pikas had been collected in three different regions (Mianyang, Aba, and Ganzi) of the Tibetan plateau. Interestingly, their virus sequences clustered according to their region of collection (Supplementary Fig. 3d), exhibiting only 65.7–94.0% RdRp amino acid identity among each region, which is indicative of a long evolutionary separation.

Sequence reads from eight of the 39 bamboo rat libraries, representing liver, colon, and duodenum tissues (Supplementary Data 2), were identified to form a new clade of arteriviruses (ArtV). This new clade is a sister to the lactate dehydrogenase-elevating virus, and exhibit 65.8–73.0% amino acid identity in their RdRp protein sequences (Fig. 4e and Supplementary Fig. 3e).

Hantaviruses were identified in three rodent and three insectivore libraries, and two *RdRp* gene sequences were assembled (Fig. 4f and Supplementary Fig. 3f). The virus (Rodent HV SC/C3 – 13.2/2021) identified from Chinese white-bellied rat had 99.6% amino acid identity of the RdRp with that of a published rodent hantavirus (MZ328260), while the shrew strain identified in *Chodsigoa hypsibia* (Insectivora HV

SC/C7 – 49.2/2022) had 86.8% identity with the shrew Altai virus (MT648514).

Bornaviruses were identified in two bat libraries and two insectivore libraries (Fig. 4g and Supplementary Fig. 3g). The mean amino acid distance for the *RdRp* gene between bat viruses and the three known genera (*Carbovirus*, *Cultervirus,* and *Orthobornavirus*) of the family *Bornaviridae* were 52.3%, 62.4%, and 56.3%, respectively. The mean amino acid distance between the insectivore viruses and the three known *Bornaviridae* genera was 50.4%, 65.2%, and 63.8%, respectively. While the amino acid distance for the three known genera in this gene region varied from 46.2% to 48.7%. Thus, we suggest that the bat and the insectivore clades represent new genera within the *Bornaviridae*.

In the case of the *Caliciviridae*, two contigs from bamboo rats clustered with a known bamboo rat sapovirus (OM480531, 85.4–98.3% amino acid identity), while two bat contigs (L33 and L152) had 94.8–95.0% amino acid identity with the bat norovirus (KJ790198). In addition, six other bat contigs (L2, L107, L117, L172, L177, and L113) fell at the base of the sapovirus phylogeny, along with viruses from humans and pigs (Fig. 4h and Supplementary Fig. 3h).

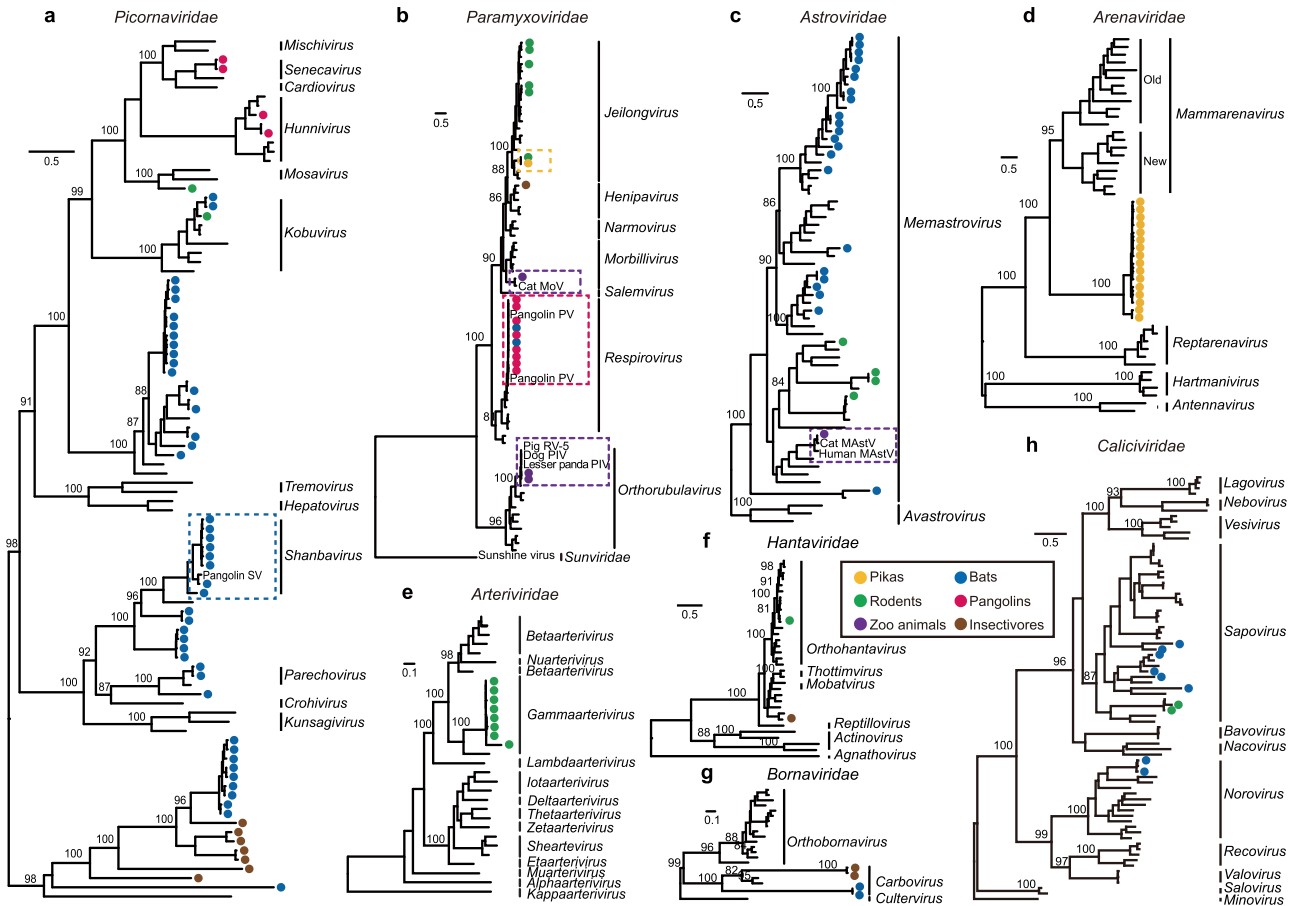

**Fig. 4 | Phylogeny of other major viral families identified in this study.**
a *Picornaviridae*, b *Paramyxoviridae*, c *Astroviridae*, d *Arenaviridae*, e *Arteriviridae*, f *Hantaviridae*, g *Bornaviridae*, and h *Caliciviridae*. Phylogeny of a *Picornaviridae*, d *Arenaviridae*, e *Arteriviridae*, f *Hantaviridae*, and g *Bornaviridae* based on the amino acid sequences of the replicase domain of RdRp. Phylogeny of

b *Paramyxoviridae*, c *Astroviridae*, and h *Caliciviridae* based on the amino acid sequences of the Large protein, the capsid protein precursor and VP1, respectively. Numbers (>80) above branches are percentage bootstrap values for the major nodes. The scale bars depict the number of amino acid substitutions per site. Detailed trees are showed in Supplementary Fig. 3.

## Circulation of DNA viruses between wildlife and domestic animals

Sequences of pseudorabies viruses (PRVs) were identified in one library from South China tigers (*Panthera tigris amoyensis*) and one library from porcupines (*Hystrix hodgsoni*). A total of five PRVs (FJ/tiger/2015, FJ/tiger/2016, FJ/tiger/2018-1, FJ/tiger/2018-2, FJ/tiger/2018-3) from South China tigers and one virus from a porcupine (FJ/porcupine/2018) were then isolated from these samples (Supplementary Fig. 4). A phylogenetic tree based on these complete genomes revealed that the five tiger sequences clustered with other porcine PRVs and belonged to genotype II (Fig. 5a). They exhibit 0.1–0.2% difference in their nucleotide sequences, and the mean genetic distance to strains of genotype II was 0.5% The porcupine strain (FJ/porcupine/2018) had a mean genetic distance of 1.4% compared to genotype I and 2.2% to genotype II. Phylogenetic trees based on the complete genome, *gB*, *gC*, and *gD* gene sequences showed that the porcupine strain clustered with genotype I, although there is a conflict with the phylogenic tree topology based on the *gE* gene, which suggested that it belonged to genotype II (Fig. 5b–e). In support of this, similarity plots of the full-length porcupine PRV genomes suggested that there were recombination events among the genotype I and II strains (Fig. 5f).

The necropsy of a tiger, which was used to isolate PRV FJ/tiger/2015, showed severe hemorrhage and congestion in the lung, as well as focal hemorrhage in the liver and kidney (Supplementary Fig. 5). The lung, liver, and spleen tissues from the tiger were used for virus

detection, pathological examination, and immunohistochemistry. Viruses were detected in all three tissues. Histopathological analysis supported severe multiorgan lesions (Supplementary Figs. 6a–c), and immunohistochemical detection of PRV antigen indicated the presence of this virus (Supplementary Fig. 6d–f).

Animal infection experiments with the PRV FJ/tiger/2015 strain showed that it can infect cats, dogs and pigs (Supplementary Table 2), which displayed typical PRV symptoms such as pruritus, anorexia, and depression. Histopathological studies showed hemorrhage and/or congestion in multiple organs, interstitial pneumonia, and non-suppurative encephalitis in infected animals (Supplementary Figs. 7–9). The PRV antigen was detected in the brain, liver, spleen, lung, and kidney of the infected animals (Supplementary Figs. 10–12), supporting active viral infection.

Circoviruses were detected in five rodent libraries and one zoo animal library (Supplementary Data 2). Interestingly, porcine circovirus 2 (PCV2) and PCV3 detected in South China tigers, were identical in their nucleotide sequences to those found in artificially domesticated wild boars that were used as live food for these tigers (Fig. 6a). In total, 168 samples from South China tigers collected between 2015 to 2022 (including 112 tissue samples from 21 dead South China tigers, 43 blood samples from 34 South China tigers, 6 anal swabs, 5 nasal swabs, and 2 fecal samples) were further examined for the presence of these viruses. qPCR showed that 68 of the 168 samples (40.5%) were positive for PCV2, while 103 (61.3%) were positive for PCV3, with 47 samples (28.0%) being positive for both

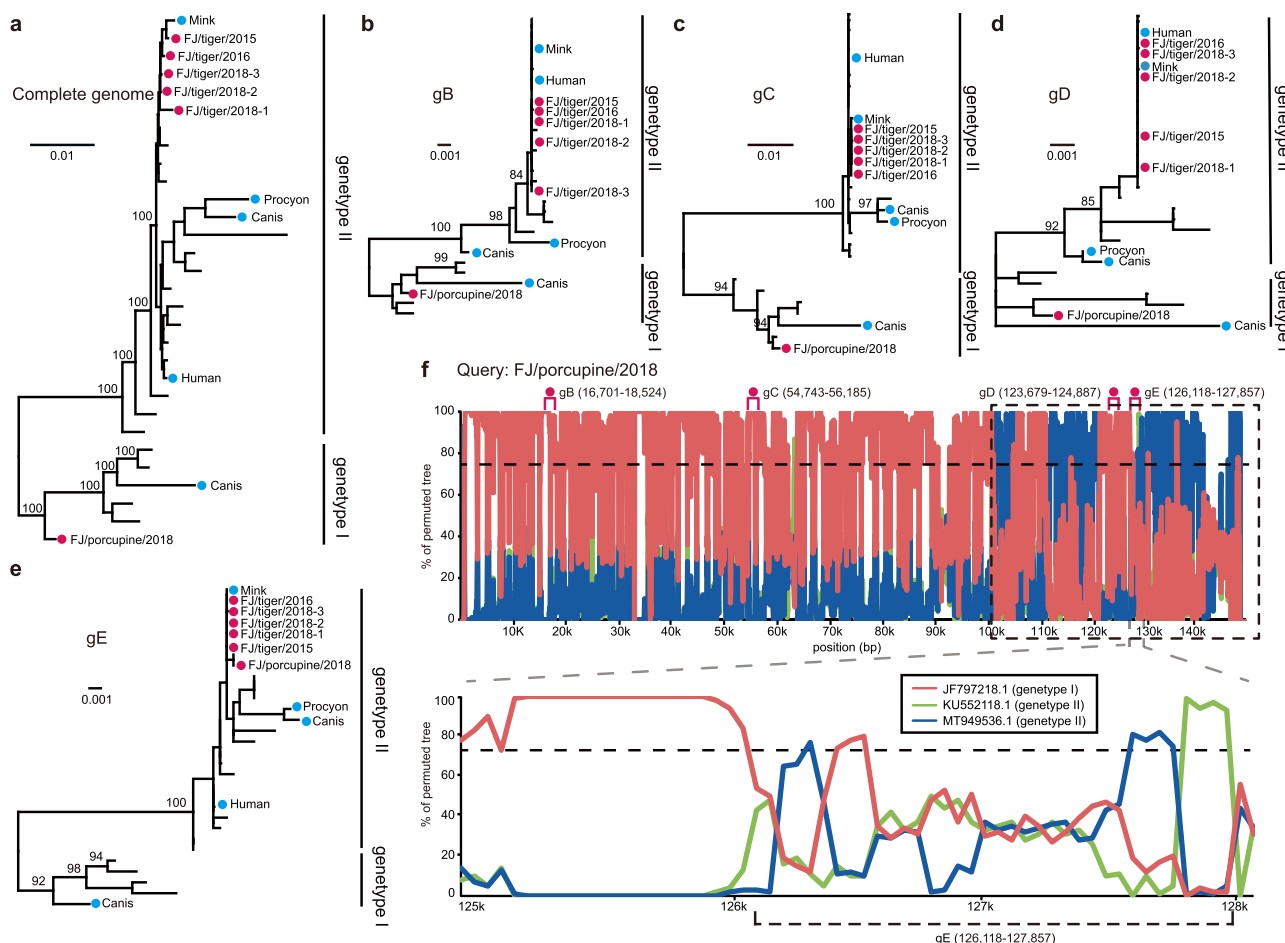

**Fig. 5 | Phylogenetic history and recombination in pseudorabies viruses (PRVs).** The phylogenetic trees were estimated based on **a** full genome sequences, and the **b** *gB*, (**c**) *gC*, **d** *gD*, and **e** *gE* genes, utilizing the best-fit nucleotide substitution model obtained by IQ-TREE (Minh et al., 2020). All trees were midpoint rooted. Numbers (>80) above branches are percentage bootstrap values for the major nodes. The scale bar represents the number of substitutions per site. Red circles indicate the six tiger- and porcupine-isolated PRVs generated in this study. **f** Similarity plot of the full-length PRV genome of porcupine PRV against sequences of a genotype I strain (JF797218.1) and two genotype II strains (KU552118.1 and MT949536.1). The parameters for the similarity plots are: window, 200 bp; step, 50 bp.

PCV2 and PCV3. In addition, 31 of 49 (63.3%) tissue or blood samples from wild boars that were used in the training of these tigers for release into the wild were positive for PCV2, 19 samples (38.8%) were positive for PCV3, and 17 samples (34.7%) were positive for both PCV2 and PCV3.

Parvovirus sequences were identified in 23, 16, 22, and 2 libraries from bats, pangolins, rodents, and zoo animals, respectively. Phylogenetic analysis showed that the pangolin parvoviruses formed a sister clade to porcine parvovirus 4, seven bat parvoviruses clustered together and were a sister clade to other published bat parvoviruses (Fig. 6b). The pangolin parvovirus in our study had three predicted ORFs, which is different from the published pangolin P229T/2018 strain (Fig. 6c). The lion parvovirus was identified and isolated from a lion that had watery/hemorrhagic diarrhea. Its genome showed 99.2% nucleotide identity with feline panleukopenia virus (FPV). Animal infection revealed that this virus can infect cats. Shedding of the virus was detected in anal swabs from three cats at 2–5 dpi. However, these cats did not show any typical clinical symptoms of feline panleukopenia, such as depression, anorexia, vomiting, and watery/haemorrhagic diarrhea.

## Discussion

New or recurrent zoonotic infectious diseases continue to pose a serious threat to public health and global economies[25]. The

surveillance of pathogens at the wildlife-domestic animal-human interface should form the basis for the prevention and control of emerging and reemerging infectious diseases.

Bats are known as the important natural hosts for coronaviruses. In the past two decades, three coronaviruses with ancestral origins in bats have emerged in humans, including SARS-CoV, MERS-CoV, and SARS-CoV-2[26]. In this study, bats were also found to harbor the highest diversity of CoVs (Fig. 2). Despite the wide distribution of SARSr-CoVs in horseshoe bats which are well-known reservoir species of SARSr-CoVs, our study had identified a MERSr-CoV in a great evening bat, but not in other bat species in the same or nearby caves in Guizhou province. This virus was also detected in this bat species in Guangdong province[27], suggesting that it likely has a wide distribution in the region. Therefore further surveillance is needed to investigate its prevalence and evolution in bats, as well as its potential spillover to other animals or even humans. In addition, bat-derived SADS-CoV emerged in Guangdong province in 2017 leading to the deaths of 24,693 piglets[28], and with subsequently reemergence in February 2019 and May 2021 in Guangdong and Guangxi provinces[29, 30]. SADSr-CoVs were detected again in horseshoe bats collected in Guangdong province in 2021. Viral spillovers from wild to domestic animals may cause animal infectious diseases and cause huge economic losses in animal husbandry. Considering the widespread and abundant prevalence of the horseshoe bats in South China and South-east Asia, this

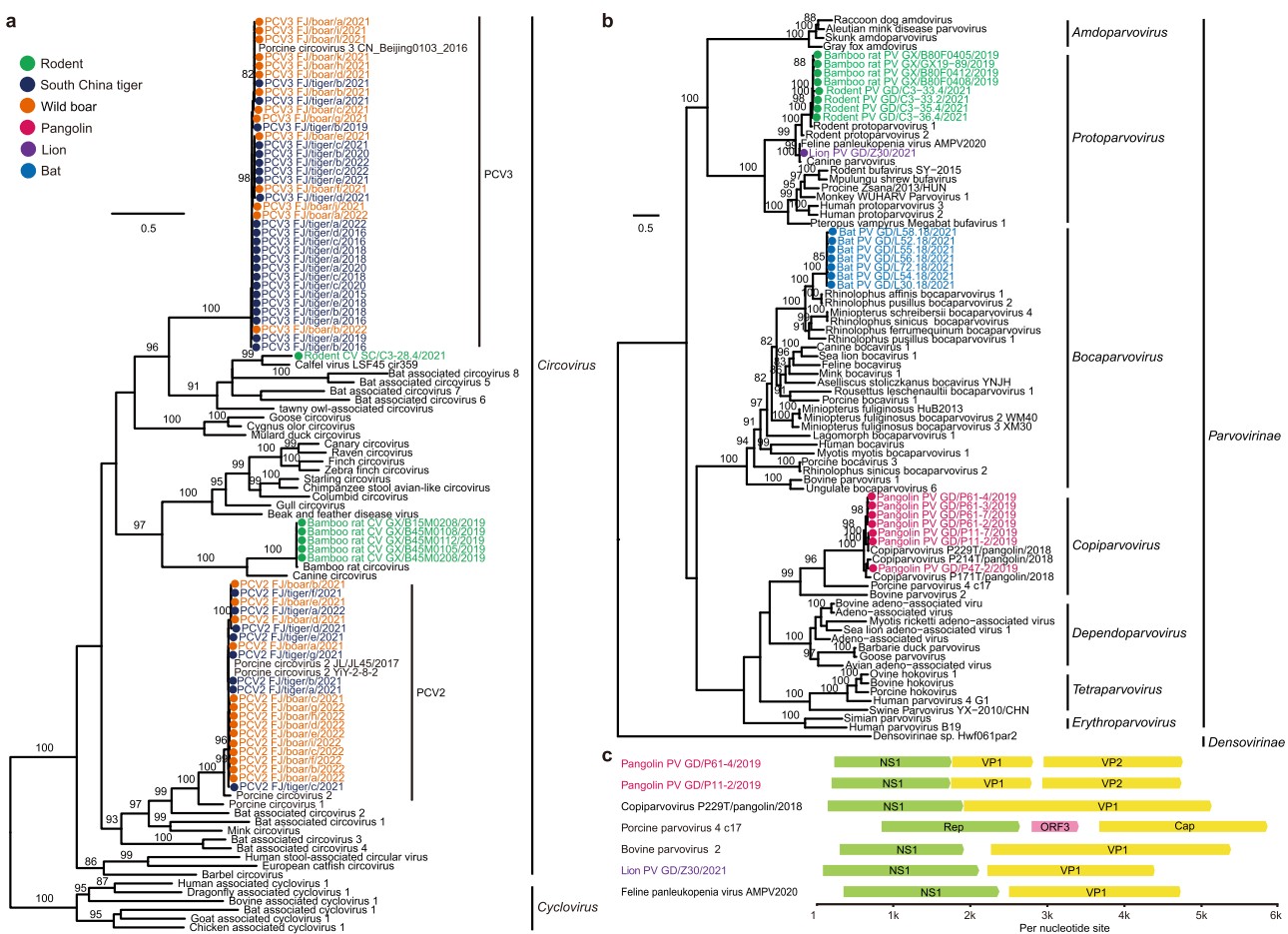

**Fig. 6 | Maximum likelihood trees of the circoviruses and parvoviruses based on the amino acid sequences of the *NS1* gene. a** Circoviruses, rooted with the cycloviruses. **b** Parvoviruses, rooted with the *Densovirinae* sp. Both phylogenetic trees were estimated with the best-fit substitutional model obtained by IQ-TREE (Minh et al., 2020) with 1000 bootstrap replicates. Numbers (>80) above branches are percentage bootstrap values for the associated nodes. **c** Genomic organization of the parvoviruses. Predicted ORFs are indicated.

emphasizes the threat of reemergence and spread of SADS-CoV in pigs. Also of note was the cluster of two rodent CoVs with a bat CoV (green box, Fig. 2a). Bats and rodents are the most important known hosts of zoonotic viruses, in part reflecting their large population sizes and wide distribution. In addition to CoVs, bats also harbor a high diversity of picornaviruses (Fig. 4a), and astroviruses (Fig. 4c). Several potentially new viruses in *Caliciviridae* (Fig. 4g) and *Parvovirinae* (Fig. 6b) were identified. Furthermore, we identified a potentially novel genus of *Bornaviridae* (Fig. 4g). Despite the intensive surveillance of bats, more surveillance work is still needed. Hantaviruses are usually rodent-borne viruses, and able to cause human hemorrhagic fever. In addition to this group of viruses identified in rodents, we found 24 viral families and 35 potentially new viruses from rodents (Supplementary Data 3). This result indicates that rodents are also important reservoirs and should be listed as a main focus of surveillance in the future.

There has been considerable interest in pangolins since they were demonstrated to be potential hosts of SARSr-CoV-2[10–12] and MERSr-CoV (HKU4-CoV-like)[13]. Of note, bat PicoV and paramyxovirus clustered closely to those from pangolins indicated potential cross-species transmission of the viruses between the two taxa (Fig. 4a, b).

In addition, we identified 11 pangolin pestiviruses. Phylogenetic analysis showed that they formed a new clade (Fig. 3), with only about 50% sequence identity with classical swine fever virus. According the classification rules of ICTV for members of the genus *Pestivirus*[31], these pangolin pestiviruses comprise a new species within this genus. Also of note was that the pestivirus obtained from a pregnant pangolin was

almost identical to that sampled from its fetus, indicating that this virus can be transmitted vertically, as has been shown for other pestiviruses[32]. Pestiviruses are among the most harmful pathogenic microorganisms to livestock[32]. Although it was originally believed that pestiviruses only infected even-toed ungulates[33], metagenomic studies have now revealed that the host range for pestiviruses includes rodents[34,35], bats[36], and cetaceans[37]. Although animal infection experiments showed that this virus cannot infect pigs and laboratory animals (mouse, rat, rabbit, and guinea pig), its potential threat to pangolins should be considered.

A new clade of Embecovirus was identified in pikas (Fig. 2a), with a 28.3% prevalence in rectum samples. It is interesting to note that the pika CoVs have a novel genome organization, in which the *HE* gene is 3' of the *S* gene (Fig. 2b), while the *HE* genes of other Embecovirus, murine hepatitis virus, rabbit HKU14-CoV and human HKU1-CoV, is 5' of the *S* gene[38]. HE is a dual-function protein, with sialic acid binding and receptor destroying activities, potentially contributes to viral entry and/or release from the cell surface via interactions with sialic acid-containing moieties[39, 40]. Murine hepatitis virus and other known rodent-borne CoVs attach to 4- or 9-O-acetylated sialosides via HE[41], while human OC43-CoV and HKU1-CoV have lost the HE lectin-binding function[42]. Whether *HE* gene and its shift of position within the genome of pika-CoV play a role in cell entry merits additional study. Alongside this β-CoV, an α-CoV was reported in this rabbit-like mammal[43] and it also harbors a new genus of arenaviruses that is a sister clade of the mammarenaviruses (Fig. 4d). Mammarenavirus are a group of

predominantly rodent-borne single-stranded RNA viruses that contain several lethal zoonotic viruses, including Lassa virus[44, 45]. Pikas have very large populations on the Tibetan plateau and are an important food source for a variety of predators, such as brown bear, Tibetan fox, saker falcon, and Eurasian sparrow hawk[46, 47]. They widely inhabit the grassland where livestock (yak, sheep, and horse) feed. Interestingly, both a α-CoV and a paramyxovirus were found to circulate among pikas and rodents collected in the same region (yellow box, Fig. 2 and Fig. 4b), suggesting that pikas could constitute virus reservoirs for the transmission to other species.

In addition, we observed two RNA viruses—paramyxovirus and astrovirus (Fig. 4b, c), and four DNA viruses—PRV, PCV2, PCV3, and FPV (Figs. 5 and 6), that were likely transmitted among domestic animals and wildlife. Notably, PRV, PCV2, and PCV3 were found to be transmitted from artificially domesticated wild boars to tigers. These boars were raised in the same zoo. Their close contact explains the transmission of the viruses between them. Swine are the natural host and reservoir of PRVs, which cause Aujeszky's disease in pigs, featuring fatal encephalitis, respiratory distress, reproductive failure, a block in the growth, and 100% mortality in newborn piglets, thereby causing catastrophic economic losses to the swine industry[48]. Surprisingly, we identified and isolated PRVs in South China tigers and a porcupine, with the former showing severe multi-organ lesions (Supplementary Figs. 5 and 6). Experimental infections showed that the tiger-isolated FJ/tiger/2015 was able to infect and cause pathological changes in multiple tissues of cats, dogs, and pigs (Supplementary Figs. 7–12). Human infections with PRV have also been reported[49–51]. It is worth noting that the five tiger-isolated PRVs belonged to genotype II, which contains isolates primarily from China and other Asian countries, while the porcupine strain was a recombinant virus between genotypes I and II (Fig. 5). Genotype I PRVs were mainly isolated from pigs from Europe and America[52]. The recombinant PRV strain suggests that genotype I virus might exist in Chinese animals, thus, additional surveys are needed to further determine its circulation within/between domestic and wildlife animals.

Also of note was that PCV2 and PCV3 had very high prevalence (40.48% and 61.31%, respectively) in captive South China tigers. Artificially domesticated wild boars are usually used for the training of these captive South China tigers back into the wild. Our results showed that these three porcine viruses (PRV, PCV2, and PCV3) jumped from pigs to tigers, causing severe tissue lesions in infected tigers (Supplementary Figs. 5 and 6). PCV2 and PCV3 are associated with porcine dermatitis and nephropathy syndrome and reproductive failure in pigs and among the most economically relevant viruses for the swine industry worldwide[53], such that their pathogenicity in tigers needs further research. Similarly, the paramyxoviruses identified in tigers were almost identical to those from pigs (purple box, Fig. 4b). These results suggest that attention should be paid to the pathogens present in the live food given to these endangered animals. In addition to these four DNA viruses, two RNA viruses (paramyxovirus and astrovirus) were found to potential circulate at the interface between wildlife in zoos and domestic animals. This not only poses a significant challenge for conservation of the endangered wildlife species, but also complicates disease control and prevention strategies in domestic animals.

The continual outbreaks caused by emerging and reemerging viruses raises considerable concern over the roles of wildlife, especially in those species that frequently contact humans and domestic animals. This study has revealed the diversity of mammalian viruses in some important mammals, and identified a series of novel genera and species of viruses with some of them having potential for cross-species transmission. More surveillance of wildlife-borne viruses, particularly at the wildlife-domestic animal-human interface, is needed to prevent outbreaks of emerging and reemerging viral diseases.

## Methods

### Animal samples
This work is part of the "fauna of Guangdong province", the "biosafety program of Guangdong province", and the "second Tibetan plateau scientific expedition and research program". Tissues, blood, feces, or pharyngeal/anal swabs from 1497 bats, 363 rodents, 58 pikas, 18 pangolins, 45 insectivorous animals, and 194 zoo animals were collected from 8 provinces of south China (details in Supplementary Data 1). Samples were stored at −80 °C until metatranscriptome sequencing.

### RNA isolation and metatranscriptome sequencing
Samples were pooled according to species, location, and tissue for subsequent RNA extraction and library construction (Supplementary Data 1). TRIzol (Invitrogen) was used for the extraction of total RNA, and rRNA was removed from the total RNA. RNA was quantified and then used for library preparation. Sequencing libraries were generated using the NEBNext Ultra Directional RNA Library Prep Kit for Illumina (NEB, USA) following the manufacturer's recommendations. In order to reduce index hoping, the strategy "prepare dual indexed libraries with unique indexes" was used, and unique dual indexes (UDI) were attached to the ends of the libraries for cross-verifying the indexes. Libraries were sequenced on the Illumina Novaseq 6000 platform (PE150, Illumina), with ~12 Gb of paired-end reads generated per sample.

### Analyses of viral-related sequences in the metatranscriptome
For each sample, low quality raw reads were filtered using Fastp (v0.23.1)[54], i.e., any reads containing adapter contamination, >10% uncertain bases, or >50% low quality bases (Phred quality <5) were discarded. After this step, clean reads were obtained, which were then aligned against the corresponding host genomes to filter out host-related reads using BWA-MEM (v0.7.17)[55]. Subsequently, host-free reads were used for de novo assembly using Megahit (v1.2.9)[56] and subjected to alignment with the non-redundant nucleotide and protein databases from NCBI using BLASTn (v2.7.1) and DIAMOND (v0.8.28.90)[57]. Viral-related contigs with e-values lower than 1e−5 were retained. The clean reads were mapped back to the assembled contigs using Bowtie2 (v2.4.4)[58] to check for coverage and depth to ensure the quality of the assembly. Taxonomic information was obtained for the top blast hit and was assigned by TaxonKit (v0.2.4)[59]. Those contigs identified as coronaviruses were further de novo assembled with coronaSPAdes (v3.15.0)[60]. Other viral genomes were obtained by mapping assembled contigs back to the reference genomes using Geneious (v2021.2.2)[61]. Coverage and depth of viral genomes were calculated with Samtools (v1.9)[62] based on the SAM files from Bowtie 2 (v2.4.4)[58]. To further improve the quality of the genome annotations, SAM files of the reads mapping to specific viral genomes were checked manually with Geneious (v2021.2.2)[61], with the ends extended as much as possible. Open reading frames (ORFs) were annotated using Geneious (v2021.2.2)[61]. The presence of viruses was further verified using RT-PCR or PCR. The PCR products were then sequenced by Sanger sequencing. We used the simplified criterion suggested by He and colleagues[25], that is, viral sequences with <80% nucleotide identity were assigned as unreported, which were then verified by detailed phylogenetic analysis.

To remove ribosomal RNA reads, sequence reads were mapped to the SILVA database (v138.1)[63] using BWA-MEM (v0.7.17)[55]. The numbers of sequencing reads that assembled into each contigs were then calculated using BWA-MEM (v0.7.17)[55]. The abundance of each virus species was estimated as the numbers of mapped reads per million total reads (RPM) in each library. Virus species with the RPM in the library lower than 1 were assumed to be false-positives due to index-hopping and were excluded from the subsequent analysis. Since the meta-transcriptomic approach is less sensitive for DNA viruses, to

confirm the presence of DNA viruses, specific qPCR was used to detect DNA viruses that might have been filtered in our data analysis pipeline, eventually resulting in the identification of PCV2 and PCV3 in tigers and boars.

## Phylogenetic and recombination analyses

Representative virus strains of each viral family used for the phylogenetic analyses were listed in Supplementary Data 6. Multiple sequence alignments were generated with a high-speed and iterative refinement method (FFT-NS-i) implemented in MAFFT (v7.407)[64]. The sequence alignments were trimmed with MEGA7 (v7.0.26)[65] and trimAI (v1.4.rev15)[66]. Maximum likelihood (ML) trees were estimated by retrieving the best-fit model of nucleotide or amino acid substitution using IQ-TREE (v2.0.3)[67] employing 1000 nonparametric bootstrap replicates. Potential recombination events and the location of possible recombinaton breakpoints in the viral genomes were detected using Simplot (v3.5.1)[68] and RDP (v4.97)[69].

## Virus isolation and genome sequencing

Pestivirus-positive tissues from pangolins, PRV-positive tissues from South China tigers and porcupines, and parvovirus-positive feces from lions were homogenized in PBS, with the suspensions harvested by centrifugation at $3000 \times g$ at 4 °C for 20 min. The suspensions were then filtered through a 0.22-μm filter and inoculated into a confluent monolayer of Vero-E6, PK-15, or F81 cells. Cells were cultured in Dulbecco's modified Eagle medium (DMEM) (Invitrogen, CA, United States) supplemented with 10% heat-inactivated fetal bovine serum (FBS) (Gibco, NY, United States), 100 μg/ml streptomycin, and 100 IU/ml penicillin at 37 °C with 5% $CO_2$ for 72 h. The cells were then freeze-thawed three times, and the virus solution was collected and inoculated into normal cells and cultured for 96 h. After 96 h of incubation, the virus solution was collected as described above for the next round of virus transfer until the cells exhibited cytopathic effects (CPEs). As the pangolin pestivirus did not exhibit CPEs, its presence was confirmed by amplifying a 643-bp fragment using RT-PCR (forward primer 5'-GCCAGACCCCACGCAACA-3' and reverse primer 5'-AGTCTCCGCATCCCCGTC-3'). PRVs were confirmed by amplifying a 500-bp fragment of the gE gene using PCR (forward primer 5'-TGGCTCTGCGTGCTGTGCTC-3' and reverse primer 5'-CATTCGTCACTTCCGGTTTC-3')[70], while the occurrence of the parvovirus was confirmed by amplifying a 681-bp fragment (forward primer 5'-AAAGAGTAGTTGTAAATAA-3' and reverse primer 5'-TATATCACCAAAGTTAGTAG-3')[71]. Next-generation Illumina HiSeq 2500 and third-generation Nanopore were employed for genome sequencing.

## Experimental infections

All experimental infections were done in biosafety level 2 animal facilities in strict accordance with the guidelines of Chinese Regulations of Laboratory Animals and Laboratory Animal-Requirements of Environment and Housing Facilities. All experimental procedures were performed in accordance with animal ethics guidelines and approved by the Animal Care and Use Committee of Longyan University (LY2022003L).

To determine the pathogenicity and the potential zoonotic potential of the tiger-isolated PRV (FJ/tiger/2015), cats, dogs, and pigs were used in the experimental infection studies. For each animal species, six 6-month-old animals that were negative to PRV were divided randomly into two groups. Animals in the experimental group were inoculated with 2 ml $10^5$ $TCID_{50}$ of the virus via hypodermic injection. Animals in the control group were inoculated with serum-free DMEM by the same dose and route. The animals were observed daily.

To determine the pathogenicity of the pangolin pestivirus (strain P42), mice, rats, guinea pigs, rabbits, and pigs were chosen for experimental infection. Six animals for each species were divided into the infected group (three animals) and the control group (three animals). The housing conditions for the mice include a temperature range of

20–24 °C with 40–60% humidity, and a 12-h light/12-h dark cycle. Animals in the infected group were subcutaneously inoculated with 1.5 ml 1.03E + 18 copies/ml of the pangolin pestivirus strain P42, while animals in the control group were inoculated with serum-free DMEM by the same dose and route. Clinical signs were recorded daily. All animals were euthanized at 7 DPI, and their heart, liver, spleen, lungs, kidneys, tonsils, and lymph nodes were collected for virus detection.

To determine the pathogenicity of the FPV isolated from a lion, four 4–6-week-old cats that were negative to FPV were used. The experimental group (three cats) was inoculated intranasally with 1 ml $10^5$ TCID50 of the virus. The animal in the control group was inoculated with serum-free DMEM by the same dose and route. Anal swabs were collected daily.

Viral DNA or RNA was isolated from the various organs (heart, liver, spleen, lungs, kidneys, tonsils, lymph nodes, and bladder) or swabs using the Viral DNA/RNA Kit, according to the manufacturer's protocol (OMEGA Bio Inc., USA, R6874-02). For RNA viruses, reverse transcription was performed using the Random Primer 6N (Takara Bio Inc., Dalian, 3801) and the PrimeScript RT Master Mix (Takara Bio Inc., Dalian, RR036A) according to the manufacturer's protocol. qPCR was performed using SYBR® Premix Ex Taq™ II (Takara Bio Inc. Dalian, RR820A) on an iQ5 iCycler detection system (Bio-Rad). Known positive and negative controls were used throughout, and all assays were performed in triplicate.

## Transmission electron microscopy (TEM)

PRV and pangolin pestivirus particles were examined using a transmission electron microscopy. For PRVs, the supernatants from cell cultures exhibiting CPEs were centrifuged at $2000 \times g$ for 15 min. The sediments were discarded, and the samples were centrifuged again at $11,000 \times g$ for 15 min to enrich for viral particles. The pellets were resuspended in 100 μl of PBS. For the pangolin pestivirus, 48 h after inoculation, cells were collected by centrifugation and fixed with electron microscope fixative for 2 h at room temperature and protected from light, followed by 1% osmium tetroxide for 2 h at room temperature. The samples were then dehydrated with an ethanol gradient, with 20 min for each step. After embedding and polymerization, 60–80 nm ultra-thin sections were prepared on copper grids. Staining was performed using 2% uranyl acetate saturated with a 2.6% lead citrate solution. The prepared grids were loaded and examined on a Hitachi HT7800 transmission electron microscope (Hitachi, TKY, Japan).

## Histopathological examination

Tissues were cut to a thickness of less than 1 cm and fixed with 4% paraformaldehyde. Tissue blocks were then sectioned into 4–5 μm thickness using a cryostat and routinely stained with hematoxylin and eosin for histopathological examination. After fixing in 4% paraformaldehyde and permeabilization with 0.25% Triton X-100, cryostat sections were double-stained for immunofluorescence assays.

## Reporting summary

Further information on research design is available in the Nature Portfolio Reporting Summary linked to this article.

## Data availability

Sequence reads generated in this study are available from the NCBI Sequence Read Archive (SRA) database under BioProject accession number PRJNA901878. The complete genome sequences of five tiger PRVs, one porcupine PRV, one lion parvovirus, one pangolin pestivirus have been deposited into GenBank under accession numbers OP727800-OP727805, OP745049, and OP868576. Assembled sequences were deposited into GenBank under accession numbers OP094593-OP094596, OP860308-OP860416, OP930871-OP930878, OP950229-OP950235, OQ236110-OQ236157, OQ297692-OQ297732, OQ348136-OQ348167, OQ363494-OQ363515, OQ363750-OQ363806,

OQ451885-OQ451890, OP094598, OP094601, OP785141, OP946514, OP971514, OQ316388, and OQ476762. Alignment files and tree files have been deposited in Zenodo (https://zenodo.org/record/7668714#.ZCmdJnZBzIU). The website of the SILVA database is http://www.arb-silva.de.

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

## Acknowledgements

This work was supported by the Guangdong Provincial Key R&D Program, (2022B1111040001 to Y.S.), Guangdong Major Project of Basic and Applied Basic Research (2020B0301030007 to Y.S.), Laboratory of Lingnan Modern Agriculture Project (NT2021007 to Y.S.), the National Natural Science Foundation of China (31872958 and 32170426 to D.G., 32022083 to B.H., and 31602048 to K.F.), the Second Tibetan Plateau Scientific Expedition and Research Program (2019QZKK0402 to D.G. and Q.Y.), the Guangdong Science and Technology Innovation Leading Talent Program (2019TX05N098 to Y.S.), the 111 Project (D20008 to Y.S.), the Department of Education of Guangdong Province (2019KZDXM004 and 2019KCXTD001 to Y.S.), the Industry-university Cooperation Project of Fujian Province (2020N5012 to K.F.), and the Major Project of Science and Technology Program of Fujian Province, China (2019NZ09005 to K.F.). We thank Lei Su and Jia Liu of ChongQing Cave Exploration Team for their assistance in sample collection.

## Author contributions

Y.S., C.T., C.D., and Y.G. designed and supervised the research. Y.S., X.Cui, W.C., K.F., W.G., B.H., J.Z, X.Li, K.L., T.C., D.G., Q.Y., J-J.Z., F.S., Y-J.W., Z.C., N.Z., K.H., and X.S. collected and processed the samples. X.Cui, X.Liang, H.W., Xing.Lu, W.G., B.H., R.C., and H.L. performed the genome assembly, sequence annotation, and phylogenetic analyses. X.Chen, X.W., J.L., P.Z., H.M., S.M., Y.L., Y.H., Xiao.Lu., and J.J. performed the Sanger sequencing and molecular detection. X.Chen, K.F., P.Z., X.W., and W.C. performed the virus isolation and animal infection experiments. Y.S., C.T., Y.G., and C.D. wrote the paper. D.M.I., D.X.L., W.G., B.H., and R.A.C. revised the paper. All authors reviewed and edited the paper.

## Competing interests

The authors declare no competing interests.

## Additional information

[1]State Key Laboratory for Animal Disease Control and Prevention, Guangdong Laboratory for Lingnan Modern Agriculture, Center for Emerging and Zoonotic Diseases, College of Veterinary Medicine, South China Agricultural University, Guangzhou 510642, China. [2]Fujian Provincial Key Laboratory for the Prevention and Control of Animal Infectious Diseases and Biotechnology, College of Life Sciences, Longyan University, Longyan 364012, China. [3]State Key Laboratory for Zoonotic Diseases, Key Laboratory for Zoonosis Research of the Ministry of Education, Institute of Zoonosis, College of Veterinary Medicine, Jilin University, Changchun 130062, China. [4]Changchun Veterinary Research Institute, Chinese Academy of Agricultural Sciences, Changchun 130122, China. [5]Guangzhou Zoo & Guangzhou Wildlife Research Center, Guangzhou 510070, China. [6]Fujian Meihuashan Institute of South China Tiger Breeding, Longyan 364201, China. [7]Integrative Microbiology Research Centre, South China Agricultural University, Guangzhou, Guangdong 510642, China. [8]Zhaoqing Branch Center of Guangdong Laboratory for Lingnan Modern Agricultural Science and Technology, Zhaoqing 526000 Guangdong, China. [9]Guangdong Provincial Wildlife Monitoring and Rescue Center, Guangzhou 510000, China. [10]Key Laboratory of Zoological Systematics and Evolution, Institute of Zoology, Chinese Academy of Sciences, Beijing 100101, China. [11]Key Laboratory of Conservation and Application in Biodiversity of South China, School of Life Sciences, Guangzhou University, Guangzhou, Guangdong 510006, China. [12]School of Mathematics, Sun Yat-sen University, Guangzhou 510275, China. [13]Department of Laboratory Medicine and Pathobiology, University of Toronto, Toronto M5S1A8, Canada. [14]Banting and Best Diabetes Centre, University of Toronto, Toronto M5S1A8, Canada. [15]Department of Avian Infectious Diseases, Shanghai Veterinary Research Institute, Chinese Academy of Agricultural Science, Shanghai 201106, China. [16]Jiangsu Co-innovation Center for Prevention and Control of Important Animal Infectious Diseases and Zoonoses, Yangzhou University, Yangzhou 225009, China. [17]Joint Influenza Research Centre (SUMC/HKU), Shantou University Medical College (SUMC), Shantou 515041, China. [18]Centre of Influenza Research, School of Public Health, The University of Hong Kong, Hong Kong, China. [19]Guangdong Provincial Key Laboratory of Zoonosis Prevention and Control, Guangzhou 510642, China. [20]These authors contributed equally: Xinyuan Cui, Kewei Fan, Xianghui Liang, Wenjie Gong, Wu Chen, Biao He. ✉e-mail: shoveldeen@shvri.ac.cn; yguan@hku.hk; changchun_tu@hotmail.com; shenyy@scau.edu.cn

