## [Peer Review File · Nature Communications]

Virus diversity, wildlife-domestic animal circulation and potential zoonotic viruses of small mammals, pangolins and zoo animalsEditorial Note: This manuscript has been previously reviewed at another journal that is not operating a transparent peer review scheme. This document only contains reviewer comments and rebuttal letters for versions considered at *Nature Communications*.

REVIEWERS' COMMENTS

Reviewer #1 (Remarks to the Author):

The authors have provided information on viral genome sequence assembled in the study, and revised the manuscript in response to my concerns. The following points should be further clarified:

1. Are the meta-transcriptomic sequencing and viral genome assembly of the samples from bats, rodents, insectivores, pikas, pangolins, and zoo animals conducted respectively or in the same batch? Are any controls included in the sequencing?
2. As previously mentioned, the SARS-CoV-2-related coronavirus sequences in Figure 2A have been published by Xiao et al (Nature 2020;583:286-289). I don't think the published data is allowed to include in analysis as a new finding in this manuscript.
3. Many previously reported viral sequences in the same animal species have been used in phylogenetic trees in this manuscript. These important publications should be cited, and authors do not need to avoid these similar studies.

Reviewer #3 (Remarks to the Author):

The revisions are suitable for this reviewer.

1. Are the meta-transcriptomic sequencing and viral genome assembly of the samples from bats, rodents, insectivores, pikas, pangolins, and zoo animals conducted respectively or in the same batch? Are any controls included in the sequencing?

Reply: As the animals used in this study were collected over a long period of time, the sequencing and genome assembly of these samples were processed in many batches. We added information of batches in Table S1 (Supplementary Data 1 in this updated manuscript).

When sequencing multiple samples in the same batch, each sample's library was assigned to a separate channel to prevent contamination and cross-talk between the libraries. Robotic arms are used throughout the sequencing process to prevent errors and contamination caused by human error. As we responded in the previous revision, the strategy of “prepare dual indexed libraries with unique indexes” was used in this study. Only specific combinations of sample indices are used, i.e. if index hopping occur and a chimeric molecule is amplified, it can be computationally identified and discarded. See Lines 444-446: “In order to reduce index hoping, the strategy “prepare dual indexed libraries with unique indexes” was used, and unique dual indexes (UDI) were attached to the ends of the libraries for cross-verifying the indexes”.

2. As previously mentioned, the SARS-CoV-2-related coronavirus sequences in Figure 2A have been published by Xiao *et al* (Nature 2020;583:286-289). I don't think the published data is allowed to include in analysis as a new finding in this manuscript.

Reply: As we responded in the previous revision, we did not use the published data as a new finding in this manuscript. Although some of the pangolins sequenced in this study are from the same batch of pangolins used in our previous study (Xiao *et al.*, Nature 2020;583:286-289), 55 meta-transcriptomes were sequenced in this study to attain their viromes rather than in Xiao *et al.* In addition to the pangolin CoV, we found many other viruses in pangolins (Figs 3 and 4). In all of the topologies (Figs 2-6), viruses assembled in this study were marked in colors to distinguish from published data. Thus the three pangolin CoVs assembled in this study were marked in

red in Fig 2A, while the two published sequences were in black..

SequenceID	Source
Pangolin CoV GD/P79-9/2019	This paper
Pangolin CoV GD/P44-9/2019	This paper
Pangolin CoV GD/M5-9/2019	This paper
Pangolin CoV GD/1/2019	From Xiao et al
Pangolin CoV GD/A22/2019	From Xiao et al

In addition, we emphasized several times that the pangolin CoV was reported in the previous studies. For example, in lines 58-59 “In addition to the reported SARSr-CoV-2 and HKU4-CoV-like viruses, picornavirus and respiroviruses also likely circulate between bats and pangolins”; lines 149-152, “Notably, three β -CoVs were detected in samples from Malayan pangolins that were from the same batch of custom confiscated animals used in our previous study¹⁰. These viruses had 99.6%-99.9% nucleotide identity with the pangolin SARSr-CoV-2 (GD1)¹⁰, and also have a close relationship with SARS-CoV-2”; Lines 353-354, “There has been considerable interest in pangolins since they were demonstrated to be potential hosts of SARSr-CoV-2¹⁰⁻¹² and MERSr-CoV (HKU4-CoV-like)³⁰”.

3. Many previously reported viral sequences in the same animal species have been used in phylogenetic trees in this manuscript. These important publications should be cited, and authors do not need to avoid these similar studies.

Reply: We respect the contributions of these previous studies and have cited 71 references in the main text. As we responded in the previous revision, we list the references for the reported viral sequences that were used in our phylogenetic trees in Table S8. There are 332 references. We are very sorry that it is impossible to list this many references in the main text, thus only some of them are cited in the main text, while the others are only listed in Table S8 (Supplementary Data 6 in this updated manuscript).